# Calcinosis in Alpaca Crias *(Vicugna pacos)* Due to Vitamin D Intoxication—Clinical, Laboratory and Pathological Findings with a Focus on Kidney Function

**DOI:** 10.3390/ani11082332

**Published:** 2021-08-07

**Authors:** Matthias Gerhard Wagener, Carina Helmer, Patricia Kammeyer, Sven Kleinschmidt, Teresa Maria Punsmann, Johanna Maria Meilwes, Cornelia Schwennen, Alexandra von Altrock, Mirja Wilkens, Barbara Schwert, Nicole von Keyserlingk-Eberius, Martin Ganter

**Affiliations:** 1Clinic for Swine and Small Ruminants, Forensic Medicine and Ambulatory Service, University of Veterinary Medicine Hannover, Foundation, 30173 Hannover, Germany; helmer@anicon.eu (C.H.); opunsma@gmail.com (T.M.P.); johanna.maria.meilwes@tiho-hannover.de (J.M.M.); cornelia.schwennen@tiho-hannover.de (C.S.); alexandra.von.altrock@tiho-hannover.de (A.v.A.); barbara.schwert@tiho-hannover.de (B.S.); nicole.keyserlingk-eberius@tiho-hannover.de (N.v.K.-E.); martin.ganter@tiho-hannover.de (M.G.); 2Lower Saxony State Office for Consumer Protection and Food Safety, Food and Veterinary Institute Braunschweig/Hannover, 30173 Hannover, Germany; Patricia.Kammeyer@laves.niedersachsen.de (P.K.); Sven.Kleinschmidt@laves.niedersachsen.de (S.K.); 3Institute of Animal Nutrition, Nutrition Diseases and Dietetics, Faculty of Veterinary Medicine, University of Leipzig, An den Tierkliniken 9, 04103 Leipzig, Germany; mirja.wilkens@vetmed.uni-leipzig.de

**Keywords:** South American camelid, iatrogenic intoxication, kidney failure, kidney function analysis, calcification, vitamin D

## Abstract

**Simple Summary:**

Alpacas in central Europe often suffer from vitamin D deficiency due to lower UV radiation. Many alpacas are therefore additionally treated with vitamin D_3_. As vitamin D_3_ contents can be given in either µg or IU (international units) and these units had been mixed up in a textbook, several alpacas were poisoned and given 40 times more vitamin D_3_ than intended. Toxicological levels of vitamin D result in an increase in the level of calcium and phosphate in the blood, thus calcification in various organs, leading to organ dysfunction. In this study, three affected alpaca crias were examined in more detail. Severe changes in the kidneys, lungs and liver were found. In addition, kidney function was evaluated.

**Abstract:**

Alpacas kept in Central Europe are often deficient in vitamin D_3_, which is supplemented orally or by injection by the owners or veterinarians. Vitamin D_3_ can be specified in two different units (IU and µg), which differ by a factor of 40. By mixing up these units, an overdosage can be induced. In this study, three alpaca crias were examined after vitamin D_3_ intoxication, with particular reference to kidney function. All three animals developed non-specific clinical alterations 1–2 weeks after a vitamin D_3_ overdose of approximately 40 times. Plasma of the animals revealed several alterations. The main findings were severe azotemia, hypercalcemia and hyperphosphatemia, 15 days after treatment. Kidney function analysis (endogenous creatinine clearance) in two of the crias revealed severe glomerular damage. All crias died despite intensive treatment within 23 days after vitamin D_3_ treatment. Necropsy revealed calcification in different organs, mainly the kidneys, lungs and liver. Since nine other crias in the same group were treated with comparable doses of vitamin D_3_ and no clinical signs were observed in these animals, it is concluded that individual animals show different levels of sensitivity to vitamin D_3_.

## 1. Introduction

The keeping of South American camelids (SACs)—especially alpacas (*Vicugna pacos*) and llamas (*Lama glama*)—is becoming more and more popular in Europe [1,2,3]. Due to their natural environment in the Andes, SACs are adapted to high exposure to ultraviolet (UV) radiation, which is important for vitamin D_3_ synthesis in the skin [4]. As vitamin D_3_ is a precursor of 1,25-dihydroxycholecalciferol, a hormone involved in calcium and phosphate homeostasis, deficiencies in vitamin D_3_ can lead to disorders in bone metabolism [5]. As UV radiation is lower in Central Europe compared to the natural habitat of SACs in the Andes, vitamin D_3_ deficiency is often diagnosed in SACs in Central and Northern Europe. Plasma vitamin D_3_ levels depend, among others, on pigmentation of the animals and the season [6]. Particularly in fall and winter, there is a decrease in vitamin D_3_ plasma levels in SACs [7,8]. Vitamin D_3_ deficiency is known to be responsible for low plasma calcium and phosphate levels and the development of rickets in growing camelids [6,9,10]. Being aware of this fact, many owners supplement their SACs, especially crias, with vitamin D_3_, either by feed additives or parenterally. Different vitamin D_3_ sources are also often combined. Various recommended dosages for vitamin D_3_ supplementation of SACs have been published by different authors [11,12,13]. This increases the risk of an overdosage by high vitamin D_3_ concentrations in ad libitum mineral feed or by parenteral injection. In the case of the animals in the present study, the attending veterinarian followed the recommendations of the scientific literature. Nonetheless, in the corresponding chapter of the textbook, the units (1 µg corresponds to 40 I.U.) were mixed up [14], which led to an overdosage of vitamin D_3_ by a factor of 40.

High plasma vitamin D_3_ levels were reported to induce calcinosis with calcifications of different organs in several domestic and wild mammals [15,16,17,18]. Toxic effects of excessive administration of vitamin D are thought to occur because of increased concentrations of 25-hydroxycholecalciferol; there is a high capacity of the liver for hydroxylation at position 25, while the hydroxylation at position 1, resulting in the production of 1,25-dihydroxycholecalciferol, is tightly regulated by parathyroid hormone and calcium. Pharmacological plasma concentrations of 25-hydroxycholecalciferol result in an activation of the vitamin D receptor, thus exerting effects similar to 1,25-dihydroxycholecalciferol. It has been shown, in k.o. mice lacking the enzyme that is pivotal for the hydroxylation at position 1, that 25-hydroxycholecalciferol can bind to the vitamin D receptor, although with lower affinity in comparison to 1,25-dihydroxycholecalciferol. In addition, high concentrations of vitamin D metabolites could result in more competition for the vitamin D binding protein, consequently, more free and therefore biologically active 1,25-dihydroxycholecalciferol, although total concentrations remain unaltered. The activation of the vitamin D receptor, either by 1,25-dihydroxycholecalciferol or by 25-hydroxycholecalciferol, increases not only plasma concentrations of calcium, but also of phosphate [19,20].

Hypercalcemia and hyperphosphatemia lead to calcification of various tissues. The first affected organs are the heart, the circulatory system and the kidneys [18,21]. Cases of calcinosis in alpacas causing damage to the arteries or the kidneys have been previously reported [22,23,24,25]. However, there is a lack of available data concerning the impact of vitamin D_3_ intoxication on the functional aspects of kidney parameters in SACs.

Therefore, the present study investigates severe cases of iatrogenic vitamin D_3_ intoxication in three alpaca crias, with a special focus on changes in kidney function.

## 2. Material and Methods

The medical history was obtained from three alpaca crias suffering from vitamin D intoxication. Furthermore, a clinical examination of all crias was performed in accordance with the routine protocols of the Clinic for Swine and Small Ruminants, Forensic Medicine and Ambulatory Service, University of Veterinary Medicine Hannover, Germany.

### 2.1. History of the Cases and Pre-Treatments

All huacaya alpaca crias (*n* = 12) of a herd consisting of a total of 55 alpacas were treated parenterally with 1,000,000–2,000,000 IU vitamin D_3_, depending on their bodyweight (it is not known whether it was an intramuscular or subcutaneous injection), in October 2016. Three of those crias aged 4–5 months were presented at the Clinic for Swine and Small Ruminants, Forensic Medicine and Ambulatory Service, University of Veterinary Medicine Hannover, Germany, due to weakness and non-specific symptoms 15 days after the vitamin D_3_ treatment (Table 1). One female cria died within a few hours after arrival at the clinic. The two remaining crias were examined and treated at the clinic.

#### 2.1.1. Cria 1

Cria 1, male, 4.5 months old, white hair, 25.5 kg body weight, had been treated with 2,000,000 IU vitamin D_3_ (vitamin D_3_ 1,000,000 I.E. ad us. vet., MEDISTAR Arzneimittelvertrieb GmbH, Holzwickede, Germany) with one shot injection 15 days before admission to the clinic. The owner reported general weakness, clinically apparent stridor and reduced feed intake since day 7 after the vitamin D_3_ injection. Subsequently the animal was presented to another veterinary clinic and treated there with florfenicol, several vitamins (vitamins A, B, C, D, E; vitamin D_3_, 25,000 IU, one shot, Ursovit AD_3_EC, wässrig pro inj., Serumwerk Bernburg AG, Bernburg, Germany), selenium and NSAIDs (flunixin-meglumin, metamizol) on day 7 after the vitamin D_3_ injection. Florfenicol treatment was repeated on day 9. Oral treatment with yeast and other oral supplements had been implemented since day 10 after the vitamin D_3_ injection. Despite these treatments, the cria had not shown any improvement in its general condition and was therefore admitted to the Clinic for Swine and Small Ruminants, Forensic Medicine and Ambulatory Service, University of Veterinary Medicine Hannover, Germany, at day 15 after the initial vitamin D_3_ injection. On the day of presentation, the owners reported anuria in the affected animal.

#### 2.1.2. Cria 2

Cria 2, male, 4 months old, brown hair, 19 kg body weight, had been treated with 1,500,000 IU vitamin D_3_ (vitamin D_3_ 1,000,000 I.E. ad us. vet., MEDISTAR Arzneimittelvertrieb GmbH) by one shot injection 15 days prior to admission to the clinic. The owner reported severe coughing with malodourous secretion of the nose since day 12 after the vitamin D_3_ injection on the day of presentation. There had been no veterinary pretreatment of this animal.

#### 2.1.3. Cria 3

Cria 3, female, 3 months old, white hair, 14.5 kg body weight, had been treated with 1,000,000 IU vitamin D_3_ (vitamin D_3_ 1,000,000 I.E. ad us. vet., MEDISTAR Arzneimittelvertrieb GmbH) by one shot injection 15 days prior to admission to the clinic. The owner reported general weakness, inactivity, respiratory problems, reduced feed and water intake but no fever since day 7 after the vitamin D_3_ injection. The cria had been examined by another veterinary clinic and pretreated with florfenicol, flunixin, several vitamins (vitamins A, B, C, D, E; vitamin D_3_, 25,000 IU, one shot Ursovit AD_3_EC, wässrig pro inj., Serumwerk Bernburg AG) and selenium, on that day. The treatment with florfenicol and flunixin was repeated two days later. Moreover, oral supplementation containing yeast, electrolytes and B-vitamins had been given orally since day 11 after the vitamin D_3_ injection. Due to a high amount of coccidia in a fecal sample, which had been investigated by the local veterinarian, toltracuril treatment was performed. As the animal’s general condition had not improved, oral supplementation (yeast, electrolytes and B-vitamins) was repeated at days 11 and 13 after the vitamin D_3_ injection. In addition, the animal was supplied with a parenteral intravenous drip infusion containing electrolytes, glucose, amino acids, bicarbonate, butafosfan and B-vitamins at day 13 after the vitamin D_3_ treatment. One day later, a vitamin paste containing 200,000 IU vitamin D_3_/L, amongst others, was given orally.

### 2.2. Clinical Pathology

Initial blood samples were taken from the jugular vein (EDTA, Monovette 9 mL K3E, Sarstedt AG & Co. KG, Nümbrecht, Germany; Lithium-Heparin, Monovette 9 mL LH, Sarstedt AG & Co. KG; serum, S-Monovette 9 mL Z, Sarstedt AG & Co. KG) for hematology and clinical chemistry (Table 2). Fecal samples were investigated for internal parasites and fecal occult blood. Parameters for hematology, clinical chemistry and fecal samples were analyzed in accordance with routine clinical laboratory methods [26,27]. Venous blood gas analysis was performed for all three animals using the Osmetech OPTI™ CCA Blood Gas Analyzer (Osmetech Critical Care Inc., Boston, MA, USA) to check pH and ionized calcium levels.

Plasma levels of 25-hydroxycholecalciferol in the current study were examined by a commercial laboratory (Synlab.vet GmbH, Markkleeberg, Germany) using chemiluminescent immunoassay (CLIA).

Kidney function and urine analysis, as well as electrophoresis of serum and urine, were performed for cria 1 (sampling on day 15 after the vitamin D_3_ injection) and cria 2 (sampling on day 16 after the vitamin D_3_ injection) (Table 3, Table 4, Table 5 and Table 6).

A kidney function analysis, based on endogenous creatinine clearance, referring to the methods published for horses, pigs and calves [28,29,30,31,32], was performed in accordance with the reference values for horses, due to a lack of reference values for alpacas (Table 4 and Table 5). Fractional excretion was calculated based on a free glomerular filtration of creatinine and no tubular secretion or absorption of it. Based on a constant creatinine excretion of 0.17 µmol/min/kg for horses, the fractional excretion of calcium and the other electrolytes was calculated in accordance with the methods described by Bickhardt et al. [30].

Electrophoresis of urine and serum samples from cria 1 and cria 2 was performed with the Elphoscan-MINI PLUS device (Sarstedt AG & Co. KG, Nümbrecht, Germany) (Table 6). The separation of the proteins was performed according to the manufacturer’s instructions on a cellulose acetate membrane (Elphor Cellulose-Acetat-Folien, Sarstedt AG & Co. KG, Nümbrecht, Germany), with 105 V as the separation voltage. A total of 40 µL of serum and 25 µL of the urine were used for electrophoresis. Due to the smaller amount of proteins in the urine of cria 2, this sample was concentrated. For concentrating, this urine was centrifuged for 10 min at 200 *g* (Rotafix 32, Andreas Hettich GmbH und Co. KG, Tuttlingen, Germany). The supernatant was transferred to ultrafiltration concentrators with a molecular weight cut-off of 10,000 Daltons and a capacity of 500 µL (Vivaspin 500, Sartorius StedimBiotech GmbH, Göttingen, Germany). Before the actual ultrafiltration, rinsing of the membrane filters was performed with 200 µL of distilled water to remove residual glycerol and sodium azide from the concentrators. Centrifugation was performed for 20 min at 15,000 *g* (Thermo Scientific HeraeusPico 17 Microcentrifuge, Heraeus Sepatech GmbH, Osterode, Germany).

## 3. Results

### 3.1. Clinical and Laboratory Findings

#### 3.1.1. Cria 1

The examination at the clinic revealed heart arrhythmia and a slight mucous secretion of the nose. Ultrasonography of the bladder revealed a pear-shaped, moderately filled bladder with a diameter of 5 cm. The cria showed no colic symptoms.

Hematology revealed slight anemia, granulocytopenia and band neutrophils (Table 2). Investigations of plasma samples revealed high levels of 25-hydroxycholecalciferol, uremia with azotemia, hypercalcemia, hyperphosphatemia, hyponatremia and hypoproteinemia; lactate levels were moderately increased (Table 2).

Due to anuria, high levels of creatinine and urea and colic-like symptoms, urolithiasis was suspected, so a diagnostic laparotomy was performed on the day of presentation. Urolithiasis was not confirmed and no obvious changes in the abdominal organs were visible. Hence, the reason for missing urination and colic-like symptoms remained unclear. A venous blood sample and a urinary sample by cystocentesis were taken simultaneously for kidney function testing (Table 3, Table 4 and Table 5).

After surgery, the cria was treated with amoxicillin, NSAID (flunixin-meglumin) and intravenous drip infusion with isotonic saline solution. The cria died 18 days after the vitamin D_3_ injection.

#### 3.1.2. Cria 2

The examination at the clinic revealed arrhythmia and occasional coughing.

There were no hematological alterations (Table 2). Investigations of plasma samples revealed high levels of 25-hydroxycholecalciferol, uremia with azotemia, hypercalcemia, hyperphosphatemia, hyponatremia and hypoproteinemia (Table 2). A blood gas analysis of a venous blood sample showed acidosis; ionized calcium was within the reference range. Lactate levels were also moderately increased (Table 2).

After admission to the clinic, the cria was treated for acidosis with sodium hydrogenphosphate and against bronchopneumonia with antibiotics (oxytetracyclin) and secretolytica (bromhexin). Moreover, the animal was treated with intensive intravenous drip infusion with isotonic saline solution due to acute kidney failure. Venous blood gas analysis was performed at day 16 after the vitamin D_3_ injection, showing lower ionized calcium levels than the initial blood sample and a higher pH level (Table 2). Kidney function was also evaluated on the same day (Table 3, Table 4 and Table 5). Plasma levels for urea, total calcium and phosphate had also increased (Table 2). The cria died 23 days after the vitamin D_3_ injection.

#### 3.1.3. Cria 3

The examination of the cria at the clinic revealed apathy and severe respiratory problems. Auscultation of the heart revealed no clearly separated heart beats. The cria was treated with dexamethasone and intravenous drip infusion immediately after submission but it died within three hours after arrival at the clinic.

Hematology of this animal revealed slight regenerative anemia and elevated band neutrophils (Table 2). Investigations of plasma samples revealed high levels of 25-hydroxycholecalciferol, uremia with azotemia, hypercalcemia, hyperphosphatemia, hypernatremia and hypoproteinemia (Table 2). A venous blood gas analysis showed that ionized calcium levels had decreased (Table 2).

### 3.2. Findings from Kidney Function Analysis

Urine analysis and kidney function analysis, as well as electrophoresis of serum and urine, were performed for cria 1 (sampling on day 15 after the vitamin D_3_ injection) and cria 2 (sampling on day 16 after the vitamin D_3_ injection) (Table 4, Table 5 and Table 6).

Evaluation of urine samples revealed severe proteinuria, as well as leucocytes, erythrocytes and renal tubule cells in the urine of both crias (Table 3).

Kidney function analysis revealed a decreased creatinine-clearance in both crias (Table 5), suggesting a decrease in the glomerular filtration rate (GFR) in both animals. Even if the underlying reference values had been established for horses and were not directly comparable with the situation in alpacas, the highly decreased GFR indicated a high glomerular dysfunction. Fractional excretion of calcium for both crias was within the reference range, but fractional excretion of phosphorous, sodium, potassium and water was severely increased and thereby showed insufficiency of tubular reabsorption. High γ-glutamyltransferase (GGT) activity and creatinine concentration in the urine of both crias gave further hints of cell damage to the renal tubules. Most of the examined parameters were found to have changed more dramatically in cria 1 than in cria 2 (Table 4 and Table 5).

### 3.3. Findings from Electrophoresis

Due to the high level of protein in the urine, electrophoresis of serum and urine samples was performed for both animals (Table 6). Investigations of serum samples revealed a hypoalbuminemia in both crias. Alpha-globulins and gamma-globulins in the serum were within the reference ranges in both animals, as were beta-globulins in cria 1, whereas beta-globulins in cria 2 had decreased. Investigations of urine samples revealed a severe proteinuria in cria 1. The protein to creatinine ratio in the urine of cria 1 was 0.0044 and 0.0007 in cria 2, respectively. Total urine protein in this animal was as high as plasma protein. All protein fractions (albumins, alpha-1-globulins, alpha-2-globulins, beta-globulins and gamma-globulins) could be detected in the urine of cria 1 (Table 6).

### 3.4. Findings from the Pathological Examination

After death, the animals underwent necropsy at the Food and Veterinary Institute Braunschweig/Hannover, Lower Saxony State Office for Consumer Protection and Food Safety (LAVES), Germany. For pathohistology, tissue sections were stained with hematoxylin and eosin stain (H. E. stain) and von Kossa stain.

Macroscopically, all animals showed multifocal thickening of the heart atrial muscle, which partly displayed a firm consistence. Accumulations of firm material were also detected in the epicard of cria 2 and in the pulmonary arteries of cria 3. Histological examination revealed severe atrial mineralization in all animals with mild myocardial degeneration and lympho-histiocytic, partly granulomatous myocarditis.

Moreover, the kidneys of all three animals were pale in color with a marked striation of the renal cortex (Figure 1a). Histologically, severe glomerular and tubular mineralization were visible, accompanied by glomerular and tubular degeneration (Figure 1b,c).

The liver of cria 2 additionally showed disseminated pin-point-sized gray foci scattered throughout the organ (Figure 2a). Histologically, severe mineralization with hepatocellular degeneration was detected (Figure 2b,c). In cria 1, mineralization and hepatocellular degeneration were more discreet and only visible histologically. Cria 3 only showed mild petechial hemorrhages in the liver parenchyma.

In the lungs of cria 2 and cria 3, multiple firm nodules of up to 2 cm in diameter were present, located in the main lung lobes (Figure 3a). Histologically, there was severe bronchial and alveolar mineralization, as well as thrombosis of lung vessels, in cria 2 (Figure 3b,c). Cria 1 only showed alveolar edema and emphysema.

Moreover, stomach compartment 3 of cria 2 displayed firm, grayish-white mucosal spots, which histologically turned out to be severe mineralization with purulent, partly ulcerative inflammation. Ulcerative inflammation of C3 was also found in cria 3, but without detectable mineralization.

In addition, cria 1 showed severe fibrinous peritonitis, pericarditis and epicarditis, most likely due to diagnostic laparotomy.

Von Kossa special stain (Figure 1c, Figure 2c, or Figure 3c) confirmed the presence of severe calcifications of the affected organs, especially of the heart atria and kidneys, as well as partly the liver, lungs and stomach compartment 3.

## 4. Discussion

The presented cases show a severe impact of vitamin D_3_ intoxication on the renal function in alpaca crias, which led to death after an acute kidney failure. Similar vitamin D intoxications have been reported in Airedale puppies, after an oral intake of vitamin D_3_ of 200,000–250,000 IU/kg body weight over several days [16], and also in cats, after being fed a commercial cat feed with overdosed vitamin D [37]. In human medicine, there are also reports of vitamin D intoxication after manufacturing and labeling errors of dietary supplements [38,39]. In the presented case, overdosage was the result of a mixing up of the vitamin D units (I.U. versus mg/mL) in a veterinary textbook the local vet had used as reference, which resulted in a 40 times overdosage of vitamin D_3_.

An oral overdosage of vitamin D in calves was reported to lead to Hyena disease with early calcification of epiphyseal plates [15,40]. Kidney failure due to vitamin D_3_-induced calcinosis in alpaca crias has been previously reported in North America by Gerspach et al. [22] and Jankovsky et al. [25]. In these previous cases reported by Gerspach et al., the affected crias had been supplemented with vitamin D orally, either by colostral supplements or a paste, up to a total cumulative dose of 231,000 and 500,000 IU vitamin D_3_. Calculated with the given 8.8 kg and 7.7 kg body weight that dose amounted to about 26,250 and 64,900 IU vitamin D/kg, respectively [22]. The cria in the report by Jankovsky et al. had been treated with 100,000 IU vitamin D at a body weight of 9 kg, which corresponds to a dose of about 12,000 IU/kg BW [25]. In our case, the vitamin D_3_ dose was administered by one shot injection and was approximately 79,000 IU vitamin D_3_/kg in crias 1 and 2 and 69,000 IU vitamin D3/kg in cria 3. In blood samples, the vitamin D_3_ status is assessed by measuring 25-hydroxycholecalciferol because the formation of 25-hydroxycholecalciferol from vitamin D in the liver is only loosely regulated. The plasma samples of all three examined crias showed severely increased 25-hydroxycholecalciferol concentrations above the upper measuring range (>3750 nmol/L) of the lab. Depending on the reference, the upper reference limits for SACs differ from 200 to 600 nmol/L [22,41]. Those upper reference limits for 25-hydroxycholecalciferol are much higher than the reference values from other studies for cattle, sheep, pigs, dogs and cats, which were reviewed by Fairwether et al. [42]. In humans, vitamin D toxicity occurs due to plasma concentrations of 25-hydroxycholecalciferol consistently above 400 nmol/L. The clinical manifestations result mainly from hypercalcemia and hyperphosphatemia and ectopic tissue calcification when the solubility product of these ions is exceeded [43].

Crias 1 and 3 received a second shot of vitamin D_3_ with a lower dose or as oral paste. Gerspach et al. assumed that such a second dose might be responsible for promoting higher serum vitamin D levels [22]. Our data do not give any indication of this hypothesis; cria 2 was treated only once with a high dose of vitamin D_3_ and also developed severe calcinosis. There might have been some unknown ways of additional oral vitamin D intake, which also have to be considered. In this herd, mineral feed was not supplemented by vitamins, but the mares received a special mash containing 4050 IU/kg vitamin D_3_. It cannot be excluded that the crias also ate some of this mash or additional vitamin D_3_ was transferred via the colostrum and milk. Different plants, such as *Trisetum flavescens*, can also promote calcinosis [44,45], but a botanical investigation of the pastures gave no indication of the presence of these plants on the affected farm.

Clinical chemistry of the blood samples of the three alpacas revealed increased creatinine, urea, calcium and potassium. These results are similar to those of the two crias reported by Gerspach et al. [22] and Jankovsky et al. [25] and agree with the reports of vitamin D_3_ intoxications in adult alpacas we reported earlier [24]. All crias revealed severe azotaemia; creatinine and urea levels in the plasma of cria 3 were up to 2905 µmol/L for creatinine and 140.3 mmol/L for urea. Other cases of azotemia in SACs were reported after acute kidney failure due to oleander and oak intoxications [46,47], or for an alpaca cria with bilateral renal agenesis [48]. This latter case had highly increased plasma levels of creatinine (1997 µmol/L), urea (46.4 mmol/L) and phosphate (5.8 mmol/L) [48], whereas the report of another alpaca cria with unilateral renal agenesis indicated only moderately increased levels of creatinine (up to 401 µmol/L) and urea (up to 19.9 mmol/L) [49]. These discrepancies can be explained by the enormous reserve capacity of the kidney.

However, comparable changes in the kidney do not necessarily have to result from an over-supply of vitamin D_3_ in every case. In lambs that also suffered from severe glomerular and tubular kidney damage as a result of nephrosis or toxic tubular necrosis, azotemia (creatinine up to 1273 µmol/L) and proteinuria were also detected by laboratory diagnosis. However, in contrast to the alpaca crias described here, hypocalcemia was present in these animals [50]. Similar findings were obtained in a fawn in which inflammatory changes in the kidneys were accompanied by azotemia, hyperphosphatemia and also hypocalcemia [51].

The data of the serum and urine electrophoresis demonstrate that there was a severe renal loss, mainly of albumin, due to insufficient tubular reabsorption.

In cria 1, both plasma and urine proteins were about 40 g/L, which could be explained by a complete destruction of the Bowman’s capsule by the calcification in this animal. In canine urine, an albumin percentage >41.4% and an albumin/alpha-1-globulin ratio >1.46 are an indication of glomerular proteinuria [52]. The albumin/alpha-1-globulin ratio in cria 1 was 10.17, therefore much higher than the above-mentioned limit for dogs. Smaller proteins (albumins and alpha-1-globulins) were higher in the urine than in the plasma, indicating a dramatic renal loss of those molecules. Additionally, even larger molecules, such as gamma-globulins, were also found in the urine, which reflects complete damage of the glomeruli.

Increasing numbers of destroyed nephrons induced increased urine flow and filtration in the remaining intact glomerula, but tubular reabsorption was probably not increased in an appropriate amount. Subsequently, K+, Na+, water and albumin reabsorption were reduced, resulting in a higher fractional excretion (Table 5). Particularly in cria 1, a highly increased fractional excretion of potassium was noticeable (Table 5), which can be explained by the high degree of tubular damage.

The results of the kidney function analysis, as well as the electrophoresis, have to be interpreted with caution. There were no reference values available for kidney function; therefore, the reference values for horses can only serve as an approximation. The reference values for serum proteins from Dawson et al. [33] had been evaluated for adult animals and not for crias. A previous study investigating serum proteins of camels revealed differences concerning the age of the animals. In serum of adult camels, there was a much higher amount of albumin than in camel crias [53], which has also to be taken into account when interpreting our results.

In addition to the changes in the kidneys, the necropsy also revealed high-grade changes in other organs, including prominent mineralization in the lungs and liver (Figure 1, Figure 2 and Figure 3). This is consistent with other reports of calcinosis in alpacas or other species where calcifications were found in the lungs and liver [21,22,23,24]. It can be assumed that mineralization in these organs is further advanced by acute renal failure and that no recovery can be expected from the resulting vicious circle in affected animals. Furthermore, these findings show that for an accurate diagnosis, a detailed medical history should be taken regarding the vitamin supplementation of the animals, since the damage to different organs can lead to a wide range of clinical symptoms. All three animals described here showed similar pathophysiological changes, which were evident both in the values determined by laboratory diagnostics and in the autopsy of the animals. These findings are also consistent with the descriptions of other authors. Nevertheless, it remains questionable why only some of the crias treated with vitamin D_3_ (3 out of 12 animals) became clinically ill. This was also observed in the herd described by Jankovsky et al., where previous deaths were attributed to other causes [25]. Overall, it can be concluded that, on the one hand, the number of cases of undiagnosed vitamin D intoxications in SACs is probably quite high; on the other hand, some animals seem to react more sensitively to high vitamin D doses than others. These results show that more basic research on the vitamin D metabolism in SACs is needed in Central Europe.

## 5. Limitations

As mentioned in the Discussion, this study is subject to some limitations. Although 12 crias were treated with significantly increased doses of vitamin D_3_, no further data are available on nine of the animals and the findings are based only on the owners’ statement that the animals were “clinically unremarkable”. Furthermore, suitable reference values for alpaca crias were not available for some laboratory diagnostic tests, so approximate reference values from other age groups or other species had to be used.

## 6. Conclusions

Our data show that increased doses of vitamin D_3_ can lead to calcinosis in alpaca crias, which can cause damage to various organs. As a result of mineralization of the kidneys, glomerular renal damage occurs, which leads to kidney failure. In the animals presented here, symptoms appeared after about 1–2 weeks after increased vitamin D_3_ administration. All animals died or were euthanized due to the poor prognosis about 3 weeks after initial vitamin D_3_ administration. Furthermore, the data show that great care should be taken in the dosing of vitamin D_3_ and in the reporting and interpretating of units.

## Figures and Tables

**Figure 1 animals-11-02332-f001:**
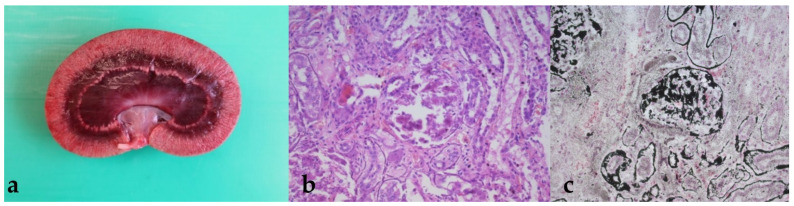
(**a**–**c**) Kidney. The cortex of the kidney appears brightened with distinct striation and disseminated grayish-white, hard foci of the cortex. Histologically, both glomerular and tubular mineralization are observed (b, H.E. stain; c, von Kossa stain).

**Figure 2 animals-11-02332-f002:**
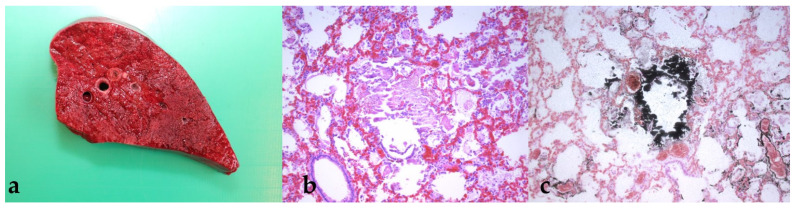
(**a**–**c**) Lung. Small bright foci in the lung parenchyma shows up histologically as mineralization of the bronchial epithelium and the interstitium (b, H.E. stain; c, von Kossa stain).

**Figure 3 animals-11-02332-f003:**
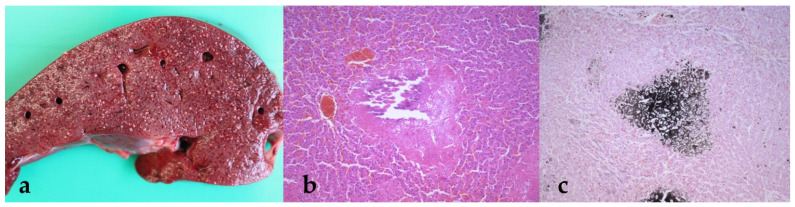
(**a**–**c**) Liver. The grayish, hard foci in the liver parenchyma show up histologically as large-area mineralization foci (b, H.E. stain; c, von Kossa stain).

**Table 1 animals-11-02332-t001:** Overview of the recommended and administered amounts of vitamin D for the three crias and the chronological sequence until the onset of symptoms and death. The recommended dose for vitamin D supplementation in alpacas is 2000 IU/kg body weight [11].

Animal	Cria 1	Cria 2	Cria 3
Sex	male	male	female
Age	4.5 months	4 months	3 months
Body weight (BW)	25.5 kg	19 kg	14.5 kg
Recommended amount of vitamin D (2.000 IU per kg BW)	51,000	38,000	29,000
Injected amount of vitamin D (IU total)	2,000,000	1,500,000	1,000,000
Injected vitamin D (IU/kg BW)	78,000	79,000	69,000
Overdosing vitamin D x	39.2 x	39.5 x	34.5 x
Time span of vitamin D injection until death	18 days	23 days	15 days
Time span of vitamin D injection until symptoms appear	7 days	12 days	7 days
Time span from onset of symptoms to death	11 days	11 days	8 days

**Table 2 animals-11-02332-t002:** Clinical chemistry of the initial and following plasma samples for cria 1 and cria 2. Analysis was performed with heparinized plasma. For the missing boxes, no data were available. Reference values derive from Dawson et al. [33] for clinical chemistry and Hengrave Burri et al. [34] for haematology, except for normoblasts, the reference values of which derive from Dawson et al. [35]. The reference value for 25-OH vitamin D was adopted from Gerspach et al. [22]. Values that differ from the references are displayed in bold.

		Cria 1		Cria 2				Cria 3
	Reference	Day 15	Day 16	Day 15	Day 16	Day 17	Day 21	Day 15
Hemoglobin (g/L)	134–166	**122**		145			**121**	**116**
PCV (L/L)	0.29–0.37	**0.25**		0.30			**0.25**	**0.23**
Leucocytes (G/L)	7.3–16.0	**3.9**		14.1			**26**	8.7
Lymphocytes (G/L)	1.4–5.9	3.22		**1.13**			**6.5**	3.22
Neutrophils (G/L)	2.9–8.0	**1.9**		**8.95**			**11.7**	3.7
Band neutrophils (G/L)	0–0.2	**0.47**		**0.35**			**5.59**	**1.18**
Eosinophils (G/L)	0.1–3.1	0.06		**3.38**			0	0.17
Basophils (G/L)	0–0.1	0.02		0.14			0	0.04
Monocytes (G/L)	0.3–3.9	0.14		0.14			2.21	0.35
Normoblasts %	0–3	2		0			0	1
25-OH vitamin D (nmol/L)	50–200	**>3750**		**>3750**				**>3750**
Creatinine (µmol/L)	88–212	**2340**	2601	**1116**	1068		**1054**	**2905**
Urea (mmol/L)	1.67–5	**93.35**	83.71	**68.94**	67.17		**128.7**	**140.3**
Calcium (total) (mmol/L)	2.09–2.54	**3.39**	**3.84**	**2.76**	2.49		**3.31**	**3.55**
Phosphate (mmol/L)	1.1–2.62	**4.81**	**5.18**	**5.32**	**5.46**		**7.18**	**7.16**
Sodium (mmol/L)	144–156	**138.3**		**142.4**				**159.4**
Potassium (mmol/L)	3.9–5.5	5.41		**6.07**				**9.04**
Protein (g/L)	58–73	**38.8**	**37.7**	**35.8**	**36.0**			**33.9**
L-lactate (mmol/L)		4.06	3.99	4.96		1.22	1.09	
D-lactate (mmol/L)		0.16	0.18	0.09		0.16	0.29	
pH		7.45	7.33	7.19	7.27	7.30		7.54

**Table 3 animals-11-02332-t003:** Urine evaluation. pH level, protein, glucose, ketone, blood and nitrite were detected by Combur 9 Test^®^, Roche Diagnostics AG, Rotkreuz, Switzerland. Specific gravity was detected by a refractometer and cells and crystals were evaluated microscopically from urine sediment after centrifugation (10 min, 200 g) at a magnification of 400x. Amounts of blood, cells and crystals are given semi-quantitatively (up to ++++). Values that differ from the references are displayed in bold.

	Cria 1	Cria 2
Physical parameters		
pH	6.5	5.5
Specific gravity	1.037	1.017
Color	yellow	light yellow
Metabolites		
Protein (g/L)	**40.5**	**6.8**
Glucose (g/L)	0	0
Ketone	-	-
Blood	**+++**	**++++**
Nitrite	-	-
Cells		
Leucozytes	**++**	**+++**
Bacteria	**++**	**+**
Erythrocytes	**++++**	**+++**
Squamous epithelial cells	-	++
Round epithelial cells	**++**	**+**
Crystals		
Ca-carbonate	++	-
Ca-oxalate	(+)	-
Tripelphosphate	-	-
Amorphous phosphate	-	-
Nonspecific crystals	+	(+)

**Table 4 animals-11-02332-t004:** Kidney function analysis. Plasma values. Cria 1, day 15 after vitamin D3 injection; Cria 2, day 16 after vitamin D3 injection. Reference values derive from Dawson et al. (2011) and were for alpacas. Values that differ from the references are displayed bolded. CK, creatine kinase; ASAT, aspartate transaminase; GLDH, glutamate dehydrogenase.

	Reference	Cria 1	Cria 2
Protein (g/L)	58–73	**41.5**	**31.5**
Albumin (g/L)	28–43	**23.1**	**13.7**
Globulin/albumin	0.9–1.7	0.8	1.3
CK (U/L)	28–144	98	53
ASAT (U/L)	125–317	**280**	100
CK/ASAT		0.4	0.5
GLDH (U/L)	3–19	12	**29**
Creatinine (µmol/L)	88–212	**2191**	**977**
Urea (mmol/L)	1.67–5	**99.38**	**58.06**
Calcium (mmol/L)	2.09–2.54	**3.36**	2.45
Phosphate (mmol/L)	1.1–2.62	**5.56**	**4.54**
Sodium (mmol/L)	144–156	**137.3**	147.7
Potassium (mmol/L)	3.9–5.5	4.15	4.65

**Table 5 animals-11-02332-t005:** Kidney function analysis. Urine values and fractional excretions (FE). Cria 1, day 15 after vitamin D3 injection; Cria 2, day 16 after vitamin D3 injection. Reference values A had been evaluated for horses [30] and not for alpacas; reference values B derive from [36] and are for llamas. Values that differ from the references are displayed in bold. GGT, γ-glutamyltransferase.

	Reference A	Reference B	Cria 1	Cria 2
Creatinine (µmol/L)			9.162	9.183
Creatinin-clearence (mL/min/kg)	1.24–2.59		**0.08**	**0.17**
Calcium (mmol/L)			2.58	0.82
FE Calcium (%)	1.3–33.2		18.36	3.56
Phosphate (mmol/L)			3.54	20.92
FE phosphate (%)	0.02–4.86		**15.23**	**49.02**
Sodium (mmol/L)			12.7	0.5
FE sodium	0.004–1.040	<1	**2.21**	0.04
Potassium (mmol/L)			170.5	53.5
FE potassium (%)	11–118	50–120	**982.49**	**122.41**
FE water (%)	0.14–2.16		**23.91**	**10.64**
GGT (U/L)		0–29	125.2	45.3
GGT (urine)/crea (urine)	0.1–0.64		**13.67**	**4.93**

**Table 6 animals-11-02332-t006:** Electrophoresis of serum and urine proteins. Reference values for serum proteins in accordance with Dawson et al. [33]. For urine proteins, no reference values for SACs were available.

	Plasma			Urine	
	Reference	Cria 1	Cria 2	Cria 1	Cria 2
Total protein (g/L)	57–72	41.5	31.5	40.5	6.8
Albumin (g/L)	29–40	19.7	12.2	29.5	3.6
Alpha-1-globulin (g/L)	1–5	1.1	1.2	2.9	2.3
Alpha-2-globulin (g/L)	3–7	3.6	3.4	3.4	0.6
Beta-globulin (g/L)	9–16	9.4	4.8	2.0	0.3
Gamma-globulin (g/L)	5–15	11.7	9.9	2.7	0

## Data Availability

All data necessary for the interpretation of this study are provided in the manuscript. The raw data evaluated for this study are located in the archives of the Clinic for Swine and Small Ruminants, Forensic Medicine and Ambulatory Service, University of Veterinary Medicine Hannover Foundation. They are not freely accessible because they are veterinary patient data, which are subject to confidentiality.

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
