# Peer review of "Calcinosis in Alpaca Crias (Vicugna pacos) Due to Vitamin D Intoxication—Clinical, Laboratory and Pathological Findings with a Focus on Kidney Function"

_animals, 2021, doi:10.3390/ani11082332_

Round 1

Reviewer 1 Report

Dear authors,

Thank you for presenting this case series regarding vitamin D toxicity in crias. 

I think this is an interesting case series; however, this manuscript requires extensive editing for spelling, grammar and syntax. You indicate that it has been through some English language editing but this manuscript is  not publishable in its current state.   

Comments: 

Line 26 (and thereafter): I suggest replacing the word 'alterations' with another term to describe pathological changes.

Line 27: Change to 'kidney function was evaluated' 

Line 35: You should be specific regarding the methodology used to evaluate kidney function (endogenous creatinine clearance for example).

Line 82: correct spacing

Line 91: 'medical history' is a preferred term

Line 112: replace 'at' with 'on'

Line 117: replace 'missing urination' with 'anuria' 

Line 124: delete 'smelling' or replace with 'malodourous' 

Line 165: again, the specific kidney function assessment methodology needs to be provided,   

Line 188: replaced 'anatomically.....' with 'underwent a necropsy' 

Line 206: for Cria 1, was there any imaging done prior to surgery? is it feasible in  a cria? 

Table 4: I am a bit confused about the description of table 4 as it appears to be two tables combined. The first part has plasma biochemistry values and the second part is urine and kidney function parameters. It would be good for the reader if you can explain why the table contains both parameters and how the fractional excretion and creatinine clearance were calculated in the methods. 

Line 285: I am not sure this is a completely valid statement as this would depend on urine concentration. A urine protein to creatinine ratio would be helpful to understand the extent of the proteinuria. 

Line 377: 'exceeded' should be replaced with 'increased' 

Line 397: typo 'of glomerular'  

Reviewer 2 Report

General Comments:

The scientific content of this document is fine, the major issues I have are to do with the structure of the article and the English content. The English is at times very clunky with incorrect phrasing which makes it difficult to read at times. Also new paragraphs are started haphazardly when not required. I also think as it currently stands the structure of the case series makes it a little difficult to follow. Please see below for more details on how I think this should be tackled.

Introduction

Line 48: This statement needs to be reworded slightly as cutaneous synthesis of vitamin D is not universal across all mammalian species.

Line 60: Implies is not the correct word to use here

Line 62: “Responsible” also incorrect word to use, change to something such as attending clinician or other

Lines 66-88: Much of this information should be moved to the start of the discussion section. In the introduction you should focus on introducing the pertinent facts surrounding your case series.

Materials and Methods:

As currently written this section is confusing. Rather than have the manuscript layout in the format you have I would suggest that you scrap the M&M and results sections and have it all combined until a title such as “Case Series” or similar. Then you would have subheadings for each individual animal and go through everything for that case. Then start the manuscript again with “Discussion”.

Line 99: Non-specific

Line 106: Change “one shot” to type of injection, i.e. “with one intramuscular injection…..” or whatever was administered

Line 108: Incorrect use of “therefore”. Change to something like “Subsequently the animal was presented to…”

Line 112-113: Do you mean it was given again 2 days later or was this the initial treatment? You need to be clearer here. Also, would be best to include dose rates if possible for all treatments and route of administration

Line 117: What is meant by “missing urination”? Do you mean the animal was anuric or it had urinated but the owners did not observe it?

Line 141: Oral supplementation with what?

Line 187-192: I understand what you are trying to do here but it does not make sense to start talking about pathology when you haven’t finished outlining the clinical course of the animals.

Line 188: Anatomically pathologically is not a correct term, you mean to say a gross necropsy was performed and tissue sections representing specific organs were collected for histopathological examination.

Results:

Line 206: Why would you think it had a urolith if you had done an ultrasound but not seen anything?

Line 208: Missing urination is not a correct medical term

Line 212-213: If the animal was in the hospital for 3 days before death you need to be a bit more specific about “intensive treatment”.

Line 216: Do you mean arrythmia?

Line 223: Treated “for” acidosis

Line 230-231: See earlier comments about intensive treatment

Discussion:

Much of this is ok.

Reviewer 3 Report

Please see my attached comments on the file. Excellent work so far!

Round 2

Reviewer 3 Report

Outstanding work and the changes to Table 3 and addition of Table 5 are spectacular!